# Vascular access devices and associated complications in paediatric critical care: A prospective cohort study

Melany Gaetani [1,2,3,4,5,6], Sarah Kleiboer[6], Randolph Kissoon[6], Kristen Middaugh[1,4,6], Christopher S. Parshuram [1,2,3,4,5,6] *

1 Child Health Evaluative Sciences, The Research Institute Hospital for Sick Children, Toronto, Ontario, Canada, 2 Departments of Paediatrics and Health Policy, Management and Evaluation, University of Toronto, Toronto, Canada, 3 Faculty of Medicine, Interdepartmental Division of Critical Care Medicine, University of Toronto, Toronto, Ontario, Canada, 4 Department of Critical Care Medicine, The Hospital for Sick Children, Toronto, Ontario, Canada, 5 Department of Paediatrics, The Hospital for Sick Children, Toronto, Ontario, Canada, 6 Center for Safety Research, Toronto, Ontario, Canada

* chris@sickkids.ca

**Data Availability Statement:** All relevant data are within the manuscript and its Supporting Information files. this has been added into the revised manuscript as suggested.

## Abstract

### Background

Though 60–80% of hospitalized patients have an intravascular device placed during hospitalization, there is a substantial risk of complication related to the placement, maintenance and removal of these devices. The objectives of this study were to describe vascular access device use, device complications and lumen dysfunction.

### Methods

An observational cohort study was conducted over a 4.5 years period, in two quaternary, university-affiliated paediatric intensive care units (ICU). Eligible patients were admitted to either the paediatric or cardiac ICU and had one or more vascular access devices in-situ at the time of enrolment. Vascular access devices were defined as any external connection directly into the circulation including peripheral and central veins, arteries or bone marrow. Consistent with practice in the studied ICUs removal of peripheral catheters was regarded as due to dysfunction or complication.

### Results

220 patients, 683 devices and 924 lumens were observed. The device complication rate was 21% and the lumen complication rate was 25%. The median duration without complication was 18 days for arterial catheters, 14 days for central venous catheters; 13 days for peripherally inserted central catheters and 4 days for peripheral intravenous catheters. On one third of all patient days, the volume of fluid administered to maintain VADs was equal to or greater than 20% of the total hourly total fluid intake.

**Funding:** The author(s) received no specific funding for this work.

**Competing interests:** The authors have declared that no competing interests exist.

## Conclusion

Approximately 1 in 5 vascular devices had one or more complications in ICU, most devices continued to be used without removal, and some complications resolved. The implications of the fluids infused to maintain device function warrants further study, as do strategies to resolve or limit the their complications in children.

## Introduction

Intravascular devices are placed in the vast majority of critically patients and up to 80% of critically ill patients have central venous devices [1–3]. Direct access to the intravascular compartment is an indispensable component of critical care: allowing delivery of life saving medications, hemodynamic monitoring and blood sampling. Complications have been reported in up to 30% of central [4–7] and up to 35% of peripheral [8–13] intravascular catheters and 20% of intra-arterial catheters [14]. Thrombosis from an activated coagulation system, the presence of static blood in incompletely flushed 'sampling' lumens, and the mixing of incompatible medications may compromise lumen function and impact the efficacy of administered therapies [9,15,16]. A recent retrospective study from our ICU showed that half of concurrent intravenous drug administrations were either incompatible or had unknown compatibility [17]. The objectives of this study were to describe the frequency, nature and timing of vascular access device (VAD) use, device complications and lumen dysfunction.

## Materials & methods

### Study design, setting and population of interest

A prospective, observational cohort study was conducted in a quaternary, university-affiliated paediatric intensive care program that cares over 2000 patient-admissions per year. Eligible patients were admitted to either the paediatric or cardiac intensive care unit (ICU) and had one or more vascular access devices in-situ at time of enrolment. Vascular access devices were defined as any external connection directly into the circulation including peripheral and central veins, arteries or bone marrow. In this center, CVCs and PICCs are most commonly inserted using ultrasound guided, modified Seldinger technique, however a significant proportion of patients have pre-existing temporary or long term vascular access devices on admission. Data were collected from a convenience sample of eligible patients identified from May 2014 to December 2018. Consent was waived given the observational nature of this study. Permission to conduct the study was provided by the Hospital for Sick Children Research Ethics Board (Approval number: 1000006271). All reporting was performed in accordance with the Strengthening the Reporting of Observational Studies in Epidemiology (STROBE) guidelines [18].

### Outcomes

The main outcomes were defined a priori, in line with current literature [7,19], and describe type and timing of device complications and lumen malfunctions. Device complications were defined as the presence of device associated thrombosis, infection, leakage, and malposition. For peripheral intravenous catheters, removal was deemed a device complication given our anecdotal direct observations of a low rate of voluntary removal of functional peripheral venous access in the intensive care units studied. Lumen malfunctions were defined as partial

or complete lumen occlusion or inability or difficulty with aspiration (Online Resource 1, S1 Table in S1 File).

## Data acquisition and management

Data was abstracted onto paper case report form and entered into a bespoke relational database (Oracle, Redwood City CA, USA). The numbers of device-days, devices, device characteristics, the associated lumens and lines and line characteristics overall and by specific device types was tabulated. Device complications were presented per device day, and per patient. Lumen dysfunction was presented per lumen day, per lumen, per device day and per patient.

Descriptive factors assessed at the patient level observations were the age, weight, presence of known pro-thrombotic condition, use of therapeutic anticoagulation, number of each of venous, arterial, peripheral access devices. Vascular access device-level observations were the type, the location, the manufacturer, the number of days since insertion, the external diameter and the length of the catheter, the number of lumens, device end point and any line level complications. The type of vascular access device was classified as arterial, peripherally inserted central venous catheters (PICC), central venous catheter (CVC) and peripheral venous (PIV). The number of days in-situ was calculated as the difference between observation date and insertion date. CVC devices included tunneled devices, dialysis catheters and umbilical venous catheters.

Lumen-level observations were lumen number, ease of infusion, ability to aspirate blood, number of stopcocks or "Y" connections in the line, patency intervention and the number lines connected, the total fluid rate running into the lumen and medications running each lumen. Lumen level complications were observed including cracks, leaks and other damage that rendered the lumen unusable. Line-level observations were tubing type used, pump type used, number of stopcocks or Y connections and the medication and or fluids running in each line, the rate of administration, the number of interventions to the line/catheter in the previous 4 hours. Estimates of fluid infused were reported as a proportion of the hourly fluid rate calculated using the 4:2:1 [20] rule administered to patients for maintenance of vascular devices. Solution types and volume administered for lumen patency in this study were in keeping with unit practice including heparin for arterial and central lines and low volumes of crystalloids administered less than or equal to 2mls/hr. TFI was calculated using the standard 4:2:1 calculation [20].

Observations were made a maximum of once per day. Each patient, device and lumen were assigned a unique identifier by the study team. Assessment of vascular access complications and lumen dysfunction was obtained by informal interview of the primary nurse. Responses were documented as yes or no, or unknown if the nurse did not know this information or if they were unavailable to the study team. Unknown information regarding device or lumen complications was assumed to be normal.

## Analysis

Demographic and clinical characteristics of devices, lumens and device and lumen complications were described using frequencies and percentages (categorical variables), means and standard deviations (SD; continuous and normally distributed variables), or medians and interquartile ranges (IQR; continuous and skewed variables). All, analyses pertaining directly to devices and complications were stratified by device. The timing of first device complication and lumen complications were described using Kaplan-Meier plots. A multistate model was used to estimate the probabilities of devices transitioning from a state of complication, no complication or removal, over time (Online Resource 1, S1 Fig in S1 File). The effects of

various covariates on the transition between states was examined. Data was analyzed using R software, version 4.1.0 (The R Foundation for Statistical Computing, Vienna, Austria). R packages are listed in the supplemental digital content (Online Resource 1, S2 Table in S1 File). No imputation for missing data was performed. Two-sided p values <0.05 were considered statistically significant.

## Results

### Patient population and vascular access device characteristics

A total of 220 patients, 683 devices and 924 lumens were observed. The median (IQR) age of children studied was 5 (1–14) months with a median (IQR) 3 (1–4) vascular access devices per patient (Table 1). The most common vascular access device among patients was a peripheral intravenous (PIV) device observed in 159 (73%) patients. One third of all PIVs placed were 22-gauge and nearly half were placed in the lower extremities. 81(39%) of central venous catheters (CVC) were placed in the right internal jugular vein with the majority (irrespective of size or length) had two lumens 148 (72%) (Fig 1). There was a total of 260 observation instances of 142 arterial individual catheters, 336 observations of 206 central venous catheters, 320 observations of 105 PICC and 323 observations of 230 PIVs. Within the included CVC devices 2% were tunneled central venous catheters, 5% haemodialysis catheters and 8% were umbilical venous catheters.

### Vascular access device complications

Device complications occurred in 258 (21%) of 1239 observed device-days (Fig 2A). Amongst the 35 (13%) arterial catheter complications thrombosis was the most common. The most common complication among CVCs and PICCs was leakage occurring in 11% and 7% respectively. PICCs had the largest proportion of infections, 5%. The most common complication to occur amongst PIVs was removal.

**Table 1. Demographic and clinical characteristics of patients observed with vascular access devices.**

|  | All Devices | No Complication | Complication (any) | P value |
|---|---|---|---|---|
| Patients, N (%) | 220 (100%) | 101 (46) | 119 (54) |  |
| Age in months, median (IQR) | 5 (1–14) | 6 (2–24) | 4 (0.75–9.5) | .01 |
| Weight (Kg), median (IQR) | 5 (3–9) | 6 (4–12) | 5 (3–7) | .01 |
| Patients with prothrombotic disease state, n (%) | 31 (14%) | 10 (10) | 21 (18) | .26 |
| Patients receiving thromboembolic prophylactic treatment n (%) | 10 (5%) | 4 (4) | 6 (5) | .60 |
| Patients receiving thromboembolic treatment n (%) | 71 (32%) | 26 (26) | 45 (38) | .08 |
| Devices |  |  |  |  |
| N, (%) | 683 (100%) | 560 (82) | 123 (18) |  |
| Number of devices per patient, median (IQR) | 3(1–4) | 2 (1–3) | 4 (2–5) | < .001 |
| Device days, median (IQR) | 3 (1–7) | 3 (1–7) | 2 (1–6) | .40 |
| Lumens |  |  |  |  |
| N, (%) | 924 (100%) | 761 (82) | 163 (18) |  |
| Per patient, median (IQR) | 4 (2–6) | 3 (2–4) | 5 (3–7) | < .001 |
| Number of drugs infused in lumen, median (IQR) | 1 (1–3) | 2 (1–3) | 1 (0–2) | < .001 |
| Infusion volume, mL/hr, median (IQR) | 2 (2–6) | 2 (2–6) | 2 (0–3) | < .001 |

Table 1 describes the demographic and clinical information of 220 patients studied. Age is the age of the patient at study enrollment. For lumen descriptions, the number of drugs and infusion volume represents the median (IQR) of each in each individual lumen. The number of drugs infused in each. There was a total of 683 devices with 924 lumens. IQR = Interquartile Range.

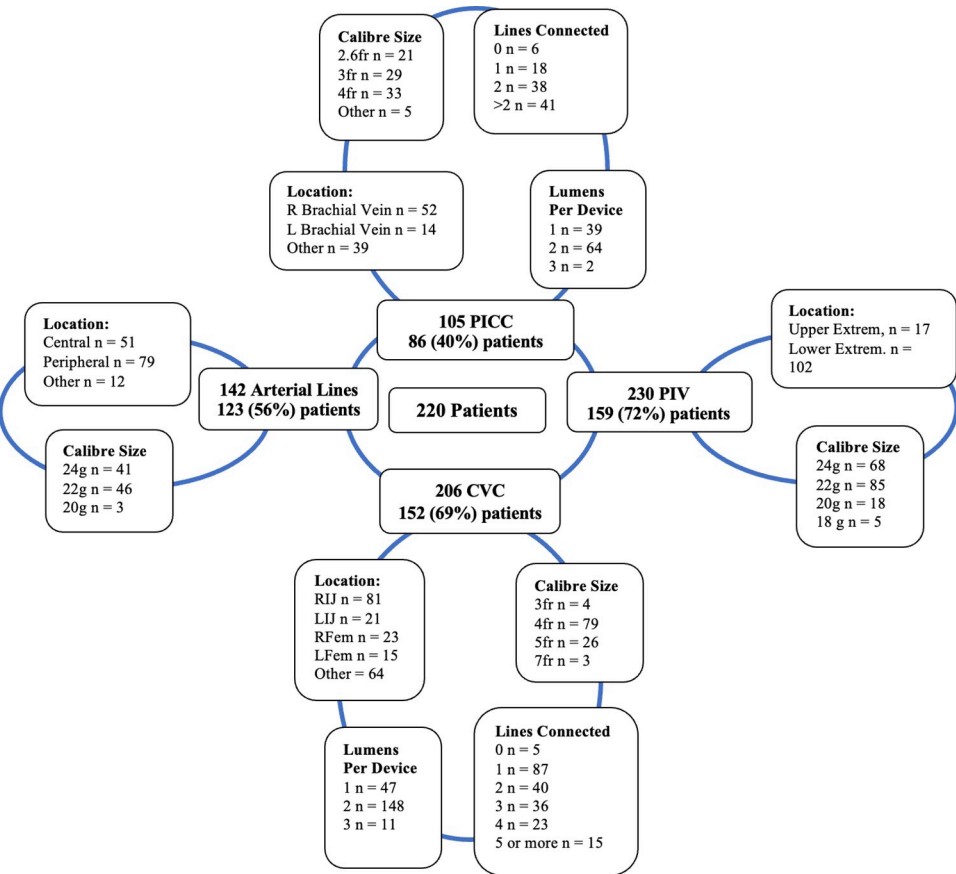

**Fig 1. Size, location and number of devices included.** The number of devices studied in the 220 studied patients. Nearly 40% of all central catheters were placed in right internal jugular vein, 50% of PICC placed in the right brachial vein and 56% of all arterial catheters are placed in peripheral vessels. ART = Arterial catheter, CVC = central venous catheter; PICC = Peripherally inserted central catheter, PIV = Peripheral Intravenous access, R = Right, L = left, IJ = internal jugular; Fem = femoral.

The overall lumen complication rate was 25% with a total of 314 dysfunctional lumens. Across all devices the most common lumen complication was the inability to aspirate blood back from one or more lumens. This occurred in 4% arterial catheters, 14% of CVCs, 13% of PICCs and 34% of PIVs (Fig 2B).

Amongst the 192 devices with one or more lumen malfunction or device complication, 78 (40%) remained in situ and were observed subsequently. The 21(20%) PICCs were used a median (IQR) 19 (7–44) days after the first complication. The 6 (4%) arterial catheters were used an additional median (IQR) 12 (8–16) days after first complication, 20 (10%) CVCs an additional median (IQR) 6 (2–9) days and 31 (13%) PIVs an additional median (IQR) 2 (1–4) days following first complication.

## Timing of device complications and lumen dysfunction

The median duration without complication was 18 days for arterial catheters, 14 days for CVCs; 13 days for PICCs and 4 days for PIVs (Fig 3). In the multistate model, once a complication had occurred the probability of removal of that device within 14 days was 71% for CVC. In PICCs with an identified complication, the probability of removal was 40% and of resolution was 46%. There was a high probability of arterial catheters removal (62%) within 7 days

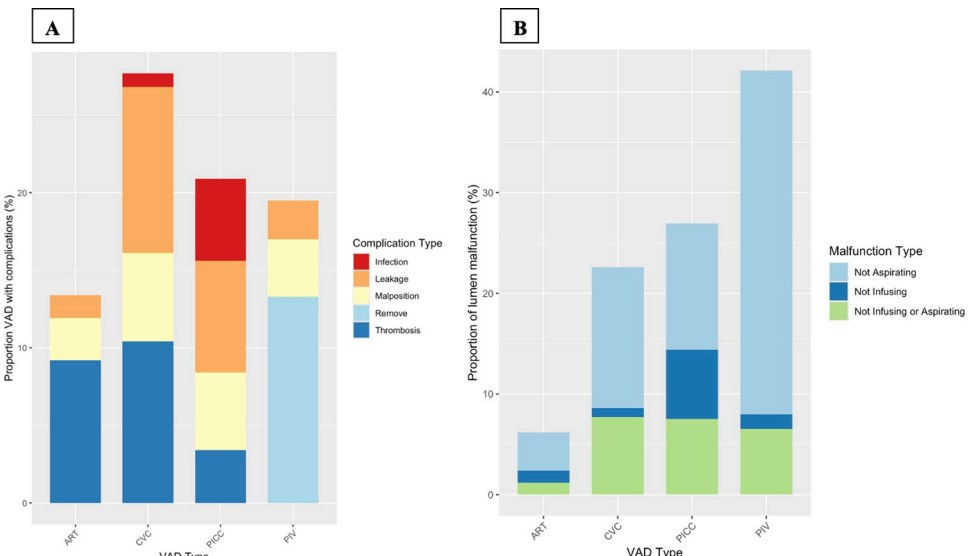

**Fig 2. Device and lumen complication rates.** The complication rate of each vascular access device by type of complication on the device level [A] and lumen level [B]. Results are displayed as a proportion of total observation days of each device. There was a total of 683 devices, observed a total of 1239 days with a total of 258 complications. Removal was only used as complication for PIV devices. The most common complication of arterial catheters was thrombosis, in both CVCs and PICCs leakage was the most common complication. All devices had a complication rate greater than 10%. There was a total of 924 lumens, observed over a total of 1239 days with a total of 314 complications. The most common malfunction across all lumen types was the inability to aspirate ranging from 3–34% of all lumens depending on the device. VAD = Vascular access device; ART = Arterial catheters, CVC = central venous catheter; PICC = Peripherally inserted central catheter, PIV = Peripheral Intravenous access.

following a complication (Online Resource 1, S3 Table in S1 File). In multistate modelling two covariates were found to be statistically significant. Smaller weight of patients was associated with lower rate of CVC removal after complication with a relative rate (95%CI) of 0.79 (0.62–0.9996), *P = .049*. Thromboembolic treatment in PICCs with a complication was found to be associated with a lower rate of removal with a relative rate (95%CI) of 0.16 (0.04–0.61), *P = .01* (Online Resource 1, S4 Table in S1 File).

## Vascular access patency and fluid volume

In 175 (80%) of patients and 418(61%) of vascular access devices fluid was administered to maintain vascular access device patency. Low volume heparin solutions were run in 152(74%) of CVCs, 139(98%) of arterial catheters and 37 (16%) of PIVs. Low volume heparin solutions for maintenance of vascular access was most commonly run in PIVs 37 (16%) and the most common solution administered for device maintenance was dextrose 5% in water with or without normal saline (0.9% NaCl) in 17 (7%) of PIVs. On one third of all patient days, the volume of fluid administered to maintain VADs was equal to or greater than 20% of the total hourly TFI calculated using the 4:2:1[20] rule (Fig 4). Among CVCs, the timing of the first lumen complication was statistically significant (*P = 0.003*) between lumens running lower volumes of fluid than those with greater volumes of fluid with a median complication free time in the low volume group of 13 days and a median complication free time of 22 days (Online Resource 1, S2 Fig in S1 File).

## Discussion

In this prospective study 220 patients, 683 vascular access devices and 924 lumens were observed to describe vascular access device use and complications in critically ill children. 50%

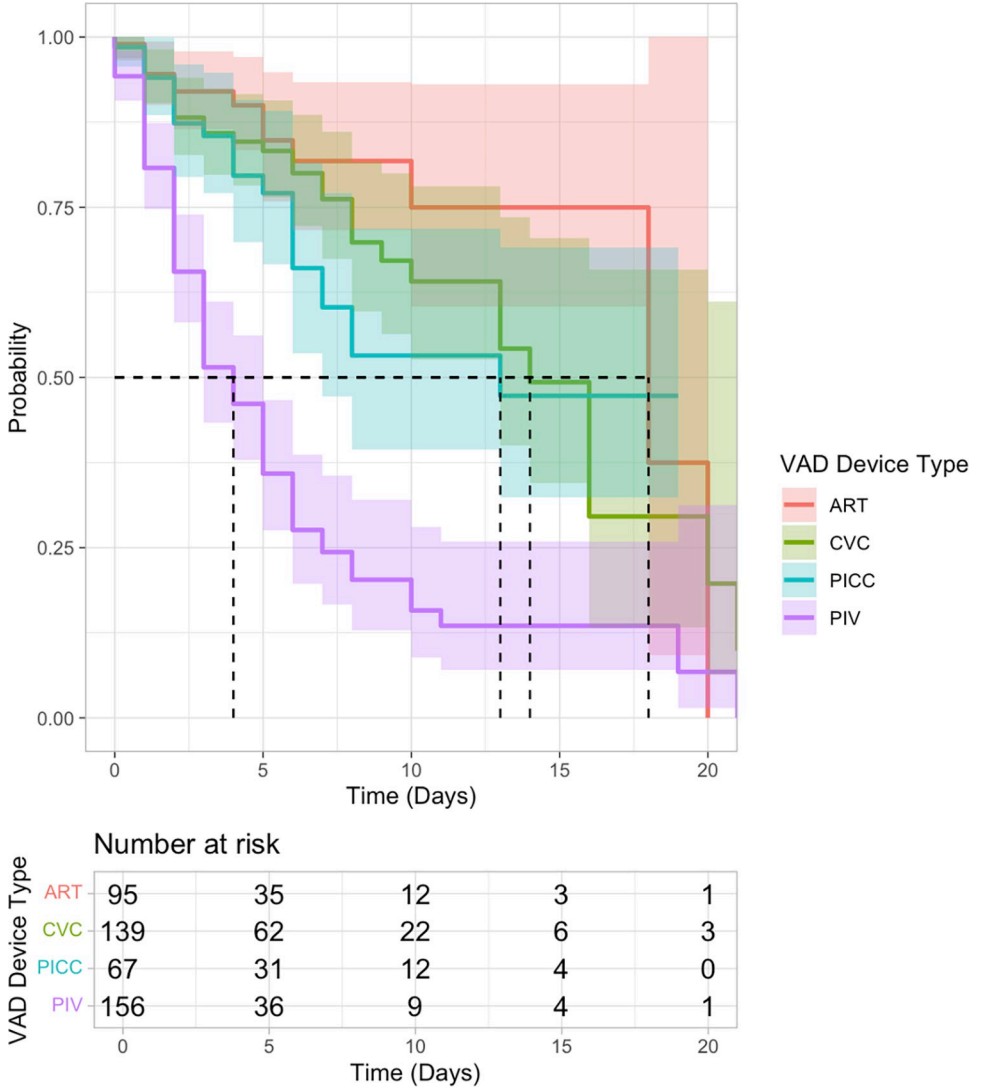

**Fig 3. Time to first complication by device.** The time to first complication for each device. The first complication could be at either the lumen or device level. The shading around each line represents the 95% confidence interval and the dashed lines represent the median duration (survival) of each device type. ART = Arterial catheters, CVC = central venous catheter; PICC = Peripherally inserted central catheter, PIV = Peripheral Intravenous access.

of patients enrolled had 3 or more vascular access devices and 54% of patients experienced complications with at least one vascular access device. There are three main findings of this work.

First, this study found that in critically ill paediatric patients, complications occurred in approximately 1 in 5 devices and 1 in 4 lumens. This rate similar to prior estimates of central vascular access device complications in various populations of children of approximately 20% [21,22]. The most common reporting of vascular access device complications are those routinely bench-marked, often pertaining to a single outcome and a type of device—commonly infections or thromboses, in central venous catheters [3,23–25]. Vascular access device complications such as mispositioning, leakage, lumen occlusion, difficulty aspirating or device removal, as reported here, are rarely collected or compared, especially across various types of vascular access devices [13,21,22,25,26]. As such, the true burden and consequences of vascular

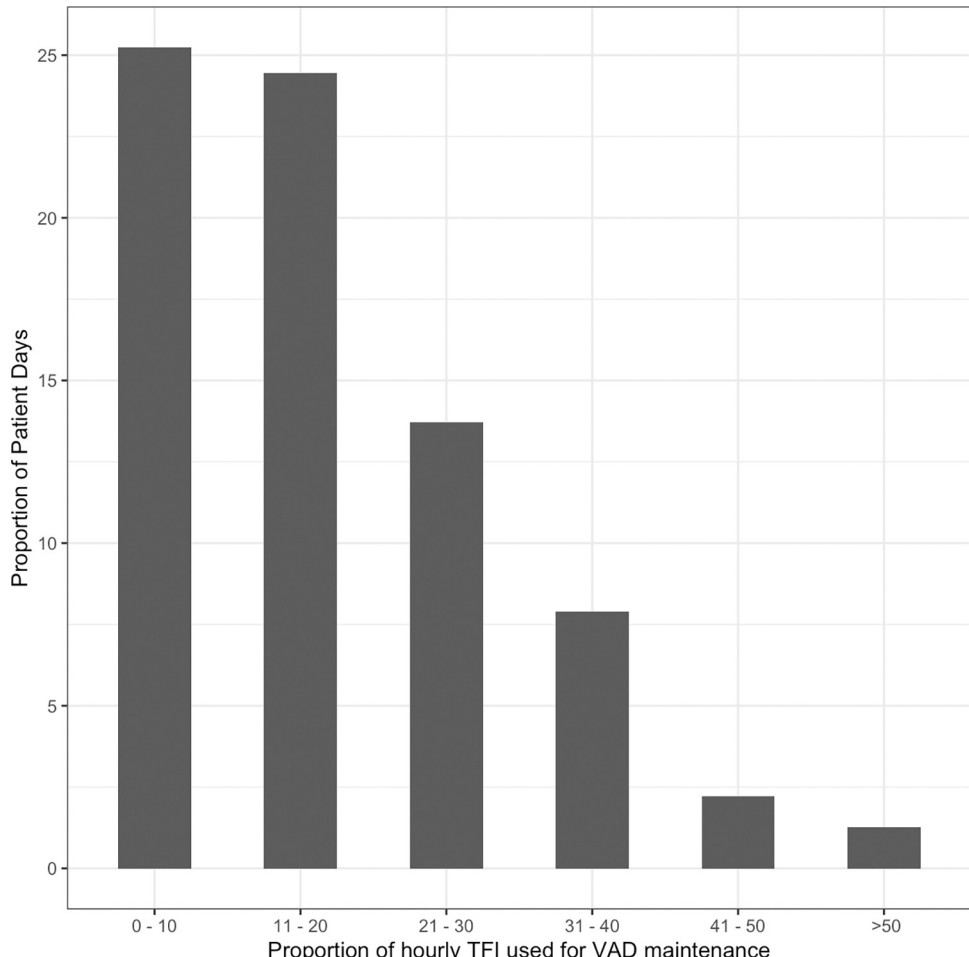

**Fig 4. Proportion of hourly fluid prescription used for vascular access maintenance.** The proportion of fluid infused as a proportion of the hourly fluid rate calculated using the 4:2:1 rule administered to patients for maintenance of vascular devices. This includes heparin for arterial and central lines and low volumes of crystalloids administered (less than or equal to 2ml/hr). TFI was calculated using the standard 4:2:1 calculation. On one third of all patient days, the volume of fluid administered to maintain VADs was equal to or greater than 20% of the total hourly TFI. TFI = total fluid intake.

access device complications for patients, families and the healthcare systems overall could be underreported.

Second, this study found that nearly half of all PICCs and CVCs will have a first complication within 2 weeks of insertion however nearly half remain in situ and will continue to be used for a median (IQR) 19 (7–44) days and 6 (2–9) days following first complication respectively. Additionally, the multistate model predicted a significant probability of complication resolution with a 47% chance of complication resolution at 30 days for PICCs and 25% probability of complication resolution in CVCs 14 days. Future work aimed at understanding bedside interventions or changes in patient physiology that contribute to the resolution of device complications could lead to the development of impactful strategies aimed at decreasing the impact of vascular access device complications, decreasing rates of premature vascular access device removal and replacement as means to improve patient outcomes.

Third, it was evident that a considerable amount of hourly fluid provided to critically ill children is used for vascular access maintenance. Prior literature provides limited justification

for this practice or related practices in paediatric patients [11], however we found significant improvement seen in the median time to lumen complications in those lumens running a larger volume. Our findings contrast with the findings of similar complication rates of PIV in a randomized trial of 12 vs 24-hourly administration of normal saline flushes in hospitalized children [27] and may reflect differences between critically ill and other inpatients. The practice of using infused fluids to maintain vascular patency may represent a potential trade-off between iatrogenic fluid overload and vascular access device function [28] and may be exacerbated by the use of multi-lumen catheters. Actively reducing the number of lumens in central venous access may reduce thrombosis, infection and cost [29,30] however, in acutely unwell patients, multi-lumen catheters are often preferred. One reason for this relates to the substantial number of unknown drug compatibilities in concurrently administered drugs in the pediatric intensive care unit [31] leading to the use of multiple lumens. Clarity around drug compatibility could result in less vascular access device complications by allowing larger volumes to flow per lumen through concurrent drug administration and allow catheters with fewer lumens to be inserted.

Finally, this work demonstrates the complexity of vascular access in critically ill children and adds to the modest body of literature, with a level of detail not previously described [3,4,32,33]. Selection of an 'ideal' device is complicated by the challenges of prospectively predicting the duration of critical illness, time-sensitive needs to provide parenteral therapies, the logistics of vascular access insertion, and the complexities of drug-drug compatibility. Vascular access device complications are frequently cited a rationale to minimize their use, to guide selection of device type and to reduce the need for replacement [4,32,34]. Current recommendations are based on a limited number of studies, expert opinion and incomplete descriptions [32,34–36], highlighting the need for a broader evidence base to inform vascular access practice. Balancing the goal of ideal function versus the opportunity costs of replacement, non-function, re-insertion and removal will continue to require thoughtful, recursive, consideration tailored to the evolving needs of each patient and the best available clinical prediction. Our findings of continued use and complication resolution reflect additional complexity clinicians must integrate into bedside decision-making about vascular access devices in critically ill paediatric patients.

## Limitations

There are several limitations to this study. First the true frequency of device complications from this study could be underestimated as devices and patients were followed intermittently and thrombotic complications were not routinely screened for in all devices and patients. Second, pathophysiological (such as severity of illness, coagulation factors) data pertaining to patient factors that could change over time and influence device complications was not gathered routinely. Third, the details of insertion and re-insertion–method, difficulty and number of attempts was not described. These may influence or be associated with complication rates. Fourth, we also did not describe the reason for removal or long-term complications of access devices. Our anecdotal experience suggests that removal of functional peripheral catheters is uncommon, and we prioritized available study resources accordingly. Future work explicitly describing reasons for removal may help identify delayed and premature removal and associated clinical consequences. Fifth, there were few tunneled devices, dialysis catheters and umbilical venous catheters limiting the robustness of estimates from these devices and the modest number of specific device types precluded meaningful analyses of specific devices and device-configurations as in previous studies [11,21]. Finally, the practices described in this single center observational study may not reflect practices at other centers–for example peripheral

venous access with butterfly needles and the use of 'midlines' may be routinely used in other ICUs.

## Conclusion

In conclusion, this prospective observational study demonstrated a wide breadth of device and lumen complications that affect critically ill children. Approximately 20% of devices will have 1 or more complications, most devices will have a first complication within 3 weeks of insertion however many devices will continue to be used and some may achieve complication resolution without removal. Future work committed to capturing a broader range of device complications and strategies that resolve or limit complications would improve current knowledge, and support improved practices for vascular access device management that contribute to improved patient outcomes.

## Supporting information

**S1 File. Contains supporting figures & tables.**
(DOCX)

## Author Contributions

**Conceptualization:** Christopher S. Parshuram.

**Data curation:** Sarah Kleiboer, Randolph Kissoon, Christopher S. Parshuram.

**Formal analysis:** Christopher S. Parshuram.

**Investigation:** Melany Gaetani.

**Methodology:** Melany Gaetani, Christopher S. Parshuram.

**Project administration:** Kristen Middaugh.

**Resources:** Christopher S. Parshuram.

**Supervision:** Christopher S. Parshuram.

**Validation:** Melany Gaetani, Christopher S. Parshuram.

**Writing – original draft:** Melany Gaetani.

**Writing – review & editing:** Melany Gaetani, Christopher S. Parshuram.

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
