## [Decision Letter · Decision Letter 0]

15 Feb 2024

PONE-D-23-38349Vascular Access Devices and Associated Complications in Paediatric Critical Care: A Prospective Cohort StudyPLOS ONE

Dear Dr. Parshuram,

Thank you for submitting your manuscript to PLOS ONE. After careful consideration, we feel that it has merit but does not fully meet PLOS ONE’s publication criteria as it currently stands. Therefore, we invite you to submit a revised version of the manuscript that addresses the points raised during the review process.

**Please revise.**

We look forward to receiving your revised manuscript.

Kind regards,

Academic Editor

PLOS ONE

Journal Requirements:

Reviewers' comments:

Reviewer's Responses to Questions

**Comments to the Author**

1. Is the manuscript technically sound, and do the data support the conclusions?

Reviewer #1: Yes

Reviewer #2: No

2. Has the statistical analysis been performed appropriately and rigorously? 

Reviewer #1: Yes

Reviewer #2: Yes

3. Have the authors made all data underlying the findings in their manuscript fully available?

Reviewer #1: Yes

Reviewer #2: Yes

4. Is the manuscript presented in an intelligible fashion and written in standard English?

Reviewer #1: Yes

Reviewer #2: Yes

5. Review Comments to the Author

Reviewer #1: Manuscript was presented as stated in the abstract. The graphs were easy to interpret and consistent with the objectives in the manuscript. The conclusion brought in the thought that as clinicians and researchers need to focus future investigation on the cause, effect and interventions necessary to promote best practice

Reviewer #2: Parshuram and colleagues provide a study investigating the occurrence of problems associated with the utilization of venous access devices in children admitted to the intensive care units at the Sick Children Hospital in Toronto.

This study includes a significant number of patients and the duration of their enrollment is sufficiently long.

However, it contains multiple deformations, mistakes, and conceptual flaws that negatively impact its quality in terms of methodology and structure/conceptual framework.

To begin with, discussing the "complications" of vascular access in such a general manner is completely inappropriate and inaccurate. Comparing peripheral venous access without providing appropriate categorization is not feasible. I would want to emphasize to the authors that peripheral venous access encompasses several methods such as a butterfly needle, a venous cannula, and a midline. This is in contrast to central and arterial access methods, and even includes intraosseous access.

This collection encompasses even arterial cannulae.

The complications associated with central venous access are distinct from those of peripheral venous access. Moreover, even within the realm of central venous access, the specific type of access being referred to is not well defined. It is important to properly categorize and differentiate between umbilical catheters, non-tunneled cuffed/non-cuffed central venous catheters (CICCs), tunneled cuffed/non-cuffed CICCs, peripherally inserted central catheters (PICCs), and ports. Failing to do so would result in a disorganized collection of these devices.

Furthermore, it is worth noting that dialysis and pheresis catheters are entirely omitted.

Moreover, there is no categorization of patients based on their age. For instance, in newborns, umbilical catheters are utilized. However, if these catheters remain in place for over 7 days or are poorly positioned, they can increase the likelihood of portal vein thrombosis. Additionally, epicutaneous-caval catheters are also employed. these examples are to say that depending on age and pathology the complications with vascular access can be different.

Combining venous and arterial access data into a single group is already a conceptual challenge in its own right. The absence of categorization based on age and pathology results in a perplexing amalgamation of data that lacks coherence.

6. PLOS authors have the option to publish the peer review history of their article (what does this mean?). If published, this will include your full peer review and any attached files.

Reviewer #1: **Yes: **Cheryl Gillette RN VA-BC CNPI

Reviewer #2: No

---

## [Author Response · Author response to Decision Letter 0]

16 Mar 2024

Response to Reviewers

The authors thank the editor and reviewers for their comments and suggestions and the opportunity to respond. Below are our itemized responses.

Journal Requirements:

Thank you. These have been updated.

Reviewers' comments:

Reviewer #1: 

Manuscript was presented as stated in the abstract. The graphs were easy to interpret and consistent with the objectives in the manuscript. The conclusion brought in the thought that as clinicians and researchers need to focus future investigation on the cause, effect and interventions necessary to promote best practice.

Thank you. 

Reviewer #2: 

Parshuram and colleagues provide a study investigating the occurrence of problems associated with the utilization of venous access devices in children admitted to the intensive care units at the Sick Children Hospital in Toronto.

This study includes a significant number of patients and the duration of their enrollment is sufficiently long.

However, it contains multiple deformations, mistakes, and conceptual flaws that negatively impact its quality in terms of methodology and structure/conceptual framework. 

[1] To begin with, discussing the "complications" of vascular access in such a general manner is completely inappropriate and inaccurate. Comparing peripheral venous access without providing appropriate categorization is not feasible. 

Thank you. We agree with the reviewer that combining venous and arterial access data into a single group is problematic. We have been careful to separate out arterial and venous access devices in all analyses, subcategorized venous access into central and peripheral venous access types and further subcategorized central venous access into directly or peripherally inserted central (PICC) venous devices. 

In response to the reviewers comments we: 

[i] Revised the methods to better represent these separations and improve clarity for readers. 

[ii] A new Table S3 has been added to more distinctly report the sub-types of central venous access devices observed. Given the modest numbers of tunneled and umbilical catheters we have resisted the temptation to conduct separate analyses of these lines. 

[iii] Conducted a sensitivity analysis. We re-ran the main study analysis including only CVL known to be percutaneously inserted temporary CVL (thus excluding peripherally inserted central catheters, umbilical and tunneled CVL). The probability of removal after a complication changed by <3%. This has been documented with the results of the main analysis and is supported by the observed frequencies of complications in table S3. 

Thank you. 

[2] I would want to emphasize to the authors that peripheral venous access encompasses several methods such as a butterfly needle, a venous cannula, and a midline. This is in contrast to central and arterial access methods, and even includes intraosseous access.

We agree with the reviewer that peripheral intravenous access represents an important subset of intravascular devices and acknowledge that readers will have experiences using different forms of peripheral intravenous access. 

On re-review of the data our anecdotal experience that butterfly needles and interosseus devices were not amongst the types of vascular access in this study has been reinforced. First, the gauge sizes recorded for the PIVs do not include available butterfly needle gauges (21, 23, 25G). Butterfly needles are used for venepuncture and are almost never used for longer term use in our anecdotal observation.

In response to these comments we have:

[i] provided more local context to better inform interpretations: highlighted observations drawn from the data and shared our local knowledge to reassure readers that peripheral venous access is unlikely to be of butterfly needle or midline type. 

[ii] revised the limitations to acknowledge that these devices were not part of the current study and that the generalizability of current results should not be assumed.

[3] This collection encompasses even arterial cannulae.

Thank you. All data analysis and figures pertaining specifically to devices and lumens differentiate arterial and venous vascular access. 

[4] The complications associated with central venous access are distinct from those of peripheral venous access. 

Thank you. We agree with the reviewer that peripheral intravenous access represents an important subset of intravascular devices and were analyzed separately from other devices.

[i] clarified the methods : to recognize the uniqueness of peripheral venous access – such that removal was considered a complication – given the relatively low rates of ‘voluntary removal’ of this type of device. 

[ii] added acknowledgment of the uniqueness of peripheral IV access by adding a citation of a systematic review of complications in adult patients 1. 

Reference:

1. Marsh N, Webster J, Ullman AJ, Mihala G, Cooke M, Chopra V, Rickard CM. Peripheral intravenous catheter non-infectious complications in adults: A systematic review and meta-analysis. J Adv Nurs. 2020 Dec;76(12):3346-3362. doi: 10.1111/jan.14565. Epub 2020 Oct 5. PMID: 33016412.)

Moreover, even within the realm of central venous access, the specific type of access being referred to is not well defined. It is important to properly categorize and differentiate between umbilical catheters, non-tunneled cuffed/non-cuffed central venous catheters (CICCs), tunneled cuffed/non-cuffed CICCs, peripherally inserted central catheters (PICCs), and ports. Failing to do so would result in a disorganized collection of these devices. Furthermore, it is worth noting that dialysis and pheresis catheters are entirely omitted.

We agree with the reviewer that these types of lines are important to be differentiated. In response we clarified the methods, added a table with the sub-types of venous access, added to results, conducted a sensitivity analysis, and acknowledged the modest numbers of sub-categories of these devices in the limitations. Please see our responses to point [1] above. 

Moreover, there is no categorization of patients based on their age. For instance, in newborns, umbilical catheters are utilized. However, if these catheters remain in place for over 7 days or are poorly positioned, they can increase the likelihood of portal vein thrombosis. Additionally, epicutaneous-caval catheters are also employed. these examples are to say that depending on age and pathology the complications with vascular access can be different.

Thank you. The reviewers point about the age of lines inserted into umbilical vessels highlights several important considerations related to these lines. 

There were several reasons we chose to focus on patient weight rather than age in the multistate model. First, these two variables are closely related to one another – and especially in younger patients – such as this cohort of patients who are largely under 14 months old. Second, from first principles and some literature, weight is good and sometimes better predictor of vessels diameter, lengths and depth (second only to height which we did not collect)1,2.

In response we have

[i] clarified scientific rationale: included the two references to better provide scientific justification for the use of weight in predictive modelling. 

[ii] included a description of age vs line type in the results (Table S3) and shown the age of unbilical lines was less than the 7 days described by the reviewer as risk-laden. 

[iii] added to the results the details of the regression analysis of risk factors for thrombosis including that we found age and weight were collinear. We have added all covariates included in the multistate model for clarity in the results. 

[iv] increased details about the types of vascular access devices: We have also listed all types of catheters include in this study in the results and a table within the supplementary data in hopes to add greater transparency and precision into the interpretation and external validity of these results. This provides readers with greater understanding about the catheters that were and were not used – including that epicutaneous-caval catheters were not used.

[v] Limitations: We acknowledge the range of devices used in this single center as a potential limitation to generalizability the lack of data pertaining to the underlying pathology of the individual patients. We agree more detailed description of patient pathologies will be a valuable component of future work. 

References:

1. López Álvarez, J. M., Pérez Quevedo, O., Santana Cabrera, L., Escot, C. R., Loro Ferrer, J. F., Lorenzo, T. R., & Limiñana Cañal, J. M. (2017). Vascular ultrasound in pediatrics: estimation of depth and diameter of jugular and femoral vessels. Journal of Ultrasound, 20, 285-292.

2. Fang, Xue-qi Master; Zhang, Hao Bachelor; Zhou, Ji-ming Master∗. Angiography in pediatric patients: Measurement and estimation of femoral vessel diameter. Medicine 99(31):p e21486, July 31, 2020. | DOI: 10.1097/MD.0000000000021486

The authors thank the reviewers for taking the time to review, for the thoughtful comments and suggestions, and thank the editor for allowing us the opportunity to respond. We believe the work has been strengthened as a consequence. Thank you 

C Parshuram and M Gaetani

For the authors.

---

## [Decision Letter · Decision Letter 1]

9 May 2024

PONE-D-23-38349R1Vascular Access Devices and Associated Complications in Paediatric Critical Care: A Prospective Cohort StudyPLOS ONE

Dear Dr. Parshuram,

Thank you for submitting your manuscript to PLOS ONE. After careful consideration, we feel that it has merit but does not fully meet PLOS ONE’s publication criteria as it currently stands. Therefore, we invite you to submit a revised version of the manuscript that addresses the points raised during the review process.

Please revise.

We look forward to receiving your revised manuscript.

Kind regards,

Academic Editor

PLOS ONE

Reviewers' comments:

Reviewer's Responses to Questions

**Comments to the Author**

1. If the authors have adequately addressed your comments raised in a previous round of review and you feel that this manuscript is now acceptable for publication, you may indicate that here to bypass the “Comments to the Author” section, enter your conflict of interest statement in the “Confidential to Editor” section, and submit your "Accept" recommendation.

Reviewer #2: (No Response)

Reviewer #3: All comments have been addressed

2. Is the manuscript technically sound, and do the data support the conclusions?

Reviewer #2: No

Reviewer #3: Partly

3. Has the statistical analysis been performed appropriately and rigorously? 

Reviewer #2: No

Reviewer #3: Yes

4. Have the authors made all data underlying the findings in their manuscript fully available?

Reviewer #2: Yes

Reviewer #3: Yes

5. Is the manuscript presented in an intelligible fashion and written in standard English?

Reviewer #2: Yes

Reviewer #3: No

6. Review Comments to the Author

Reviewer #2: After thoroughly reviewing the revision suggested by the Authors following my feedback on the initial version of the work, I found that while the revisions and accompanying explanations provided by the Authors are pertinent and sufficient, I still find it challenging to identify the coherent structure of the work. The combination of various types of venous accesses, including peripheral cannulae, arterial cannulae, central venous catheters, umbilical catheters, and dialysis catheters, along with the accompanying analysis of associated issues, appears to be burdensome and disorganized from a conceptual standpoint. Moreover, it may potentially lead to confusion and errors for readers. For certain categories such as PICC and central venous catheters, the method of insertion is not specified, which is a crucial component in the development of problems.

Reviewer #3: It's intriguing to note that vascular access utilization, a crucial aspect of Pediatric Critical Care, needs to be extensively explored in the literature. This could be due to various factors, including hospital liability. However, it's important to shift the focus from the complications associated with vascular access to its necessity, a perspective that your work can significantly contribute to.

The manuscript is a brief study of how long the catheters last until they are removed. I suggest a significant restructuring.

First, an appropriate introduction to what kind of vascular access we need: The authors can read the MAGIC (miniMAGIC) protocol and understand and better express why we need specific types of access. Ullman AJ, Bernstein SJ, Brown E, Aiyagari R, Doellman D, Faustino VS, et al. The Michigan appropriateness guide for intravenous catheters in pediatrics: miniMAGIC. Pediatrics. (2020) 145(Suppl 3):S269–84.

Then, they can review a similar study, such as the one presented: Peripheral vascular access as exclusive access mode in the pediatric intensive care unit. (Armstrong)

With a comprehensive understanding of the existing literature and the recommendations in the field, I am confident that your manuscript can be adjusted to make a significant and meaningful contribution.

The etiology of removal has to be well-defined. What is a catheter clot, a vein clot, an infection, etc.?

But again, once I see a completely revised version, more in line with what has been published in the field, I will be happy to assist. I need the authors’ help to recommend or give specific instructions.

Though 60-80% of hospitalized patients worldwide will have an intravascular device

placed during hospitalization

What is the purpose of this statement? The study is for the ICU patients, and I am sure the authors will agree that they don’t have 20-40% admissions without vascular access in their ICU. Please either remove it or provide the appropriate ICU vascular access rate.

Specific to peripheral intravenous lines, removal was also considered a device complication, given the relatively low ‘voluntary removal’ of peripheral venous access in intensive care.

What does this statement mean? Should patients stay in ICU when they don’t even need vascular access?

central venous lines(CVL) = let’s call them what they are Central venous catheters or CVCs

The most common complication to occur amongst PIVs was removal. Is removal a complication? Are you thinking about dislodgement?

7. PLOS authors have the option to publish the peer review history of their article (what does this mean?). If published, this will include your full peer review and any attached files.

Reviewer #2: No

Reviewer #3: **Yes: **Thomas Spentzas

---

## [Author Response · Author response to Decision Letter 1]

10 Jul 2024

Response to Reviewers

The authors wish to thank the reviewers for their thoughtful comments and suggestions, and to the editors for the providing us the opportunity to respond. We believe that the manuscript has improved as a consequence. 

Please find our edits and additions to each addressable point in the response outlined below. 

Thank you. 

Sincerely,

Melany Gaetani, Sarah Kleiboer, Randolph Kissoon, Kristen Middaugh, Christopher S Parshuram.

PONE-D-23-38349R1

Vascular Access Devices and Associated Complications in Paediatric Critical Care: A Prospective Cohort Study

PLOS ONE

Reviewers' comments:

Reviewer's Responses to Questions

Comments to the Author

1. If the authors have adequately addressed your comments raised in a previous round of review and you feel that this manuscript is now acceptable for publication, you may indicate that here to bypass the “Comments to the Author” section, enter your conflict of interest statement in the “Confidential to Editor” section, and submit your "Accept" recommendation.

Reviewer #2: (No Response)

Reviewer #3: All comments have been addressed

Thank you.

2. Is the manuscript technically sound, and do the data support the conclusions?

Reviewer #2: No

Reviewer #3: Partly

Thank you.

3. Has the statistical analysis been performed appropriately and rigorously?

Reviewer #2: No

Reviewer #3: Yes

Thank you. Please see responses to specific comments below. 

4. Have the authors made all data underlying the findings in their manuscript fully available?

Reviewer #2: Yes

Reviewer #3: Yes

5. Is the manuscript presented in an intelligible fashion and written in standard English?

Reviewer #2: Yes

Reviewer #3: No

Thank you. We believe the written English is sensible from grammatical and syntactic standpoints. We are happy to revise further in copy-editing or at earlier stage to improve clarity of the messages. 

6. Review Comments to the Author

Reviewer #2: After thoroughly reviewing the revision suggested by the Authors following my feedback on the initial version of the work, I found that while the revisions and accompanying explanations provided by the Authors are pertinent and sufficient, I still find it challenging to identify the coherent structure of the work. The combination of various types of venous accesses, including peripheral cannulae, arterial cannulae, central venous catheters, umbilical catheters, and dialysis catheters, along with the accompanying analysis of associated issues, appears to be burdensome and disorganized from a conceptual standpoint. Moreover, it may potentially lead to confusion and errors for readers. 

We thank the reviewer for their ongoing engagement in the review process and acknowledge that the various combinations of vascular access devices in critical care and their real-time management are complex. 

The overarching goal of the work was (and is) to categorize and evaluate this complexity without the sacrifice of clinically relevant information that often accompanies ‘simplification’. We believe this is one feature that differentiates this work from other work in the field and reflects nuances of devices considered by clinicians in practice at the bedside. 

In response to this over-arching suggestion we have: 

[1] revised the nomenclature to be more consistent (following reviewer 2 suggestion)

[2] Revised supplementary table 1 to include a legend explaining context of use. 

[3] revised the discussion to include acknowledgement of the complexity and the need for greater understanding – first through observational studies, then interventional work – to inform best practice with objective evidence.

For certain categories such as PICC and central venous catheters, the method of insertion is not specified, which is a crucial component in the development of problems.

Thank you. We did not abstract this – due to limited availability. 

In response we have

[1] clarified these points in the methods.

“In this center, CVCs and PICCs are most commonly inserted using ultrasound guided Seldinger technique, however a significant proportion of patients have pre-existing temporary or long term vascular access devices on admission.”

[2] emphasized this as in the revised limitations section. 

Thank you. 

Reviewer #3: It's intriguing to note that vascular access utilization, a crucial aspect of Pediatric Critical Care, needs to be extensively explored in the literature. This could be due to various factors, including hospital liability. However, it's important to shift the focus from the complications associated with vascular access to its necessity, a perspective that your work can significantly contribute to. The manuscript is a brief study of how long the catheters last until they are removed. I suggest a significant restructuring.

[1] First, an appropriate introduction to what kind of vascular access we need: The authors can read the MAGIC (miniMAGIC) protocol and understand and better express why we need specific types of access. Ullman AJ, Bernstein SJ, Brown E, Aiyagari R, Doellman D, Faustino VS, et al. The Michigan appropriateness guide for intravenous catheters in pediatrics: miniMAGIC. Pediatrics. (2020) 145(Suppl 3):S269–84. Then, they can review a similar study, such as the one presented: Peripheral vascular access as exclusive access mode in the pediatric intensive care unit. (Armstrong). With a comprehensive understanding of the existing literature and the recommendations in the field, I am confident that your manuscript can be adjusted to make a significant and meaningful contribution.

Thank you. As clinicians with over 30 years of critical care experience in busy quaternary PICUs, we appreciate the sentiments expressed by this reviewer. Our research program includes considerations of the nature of parenteral therapies, intravascular fluids, drug-drug compatibility and this work evaluating the nature of vascular access devices, their function and the associated patient-related complications.

We have incorporated descriptions in the cited work in the revised manuscript. Thank you. 

“…, this work demonstrates the complexity of vascular access in critically ill children and adds to the modest body of literature, with a level of detail not previously described. Selection of an ‘ideal’ device is complicated by the challenges of prospectively predicting the duration of critical illness, time-sensitive needs to provide parenteral therapies, the logistics of vascular access insertion, and the complexities of drug-drug compatibility. Vascular access device complications are frequently cited a rationale to minimize their use, to guide selection of device type and to reduce the need for replacement. Current recommendations are based on a limited number of studies, expert opinion and incomplete descriptions, highlighting the need for a broader evidence base to inform vascular access practice. Balancing the goal of ideal function versus the opportunity costs of replacement, non-function, re-insertion or removal will continue to require thoughtful, recursive, consideration tailored to the evolving needs of each patient and the best available clinical prediction. Our findings of continued use and complication resolution reflect additional complexity clinicians must integrate into bedside decision-making about vascular access devices in critically ill paediatric patients.”

[2] The etiology of removal has to be well-defined. 

We agree with the reviewer that the etiology of removal is an important component of vascular devices however the objectives of this study were to describe the frequency, nature and timing of vascular access device (VAD) use, device complications and lumen dysfunction. Removal was a secondary consideration. 

“Fourth, we also did not describe the reason for removal or long-term complications of access devices. The tracking of reasons for removal were not feasible with available study resources. Future work to delineate reasons for removal may help identify delayed and premature removal and associated clinical consequences.” 

[3] What is a catheter clot, a vein clot, an infection, etc.?

Thank you. We agree detailed descriptions of interpretations of ‘clot’ are important and can be found in the supplementary material in S1 Table: outcome definitions. These include functional obstruction of a lumen, and vessel associated thrombosis – with ultrasound confirmed vascular obstruction. 

[4] But again, once I see a completely revised version, more in line with what has been published in the field, I will be happy to assist. I need the authors’ help to recommend or give specific instructions.

Thank you. We leave the decision about the need for further revisions to the manuscript with the editors, and here reflect that the publication of work that is ‘in line’ with previous authors thinking may limit diversity of perspectives and inhibit advances in understanding. 

[5] Though 60-80% of hospitalized patients worldwide will have an intravascular device placed during hospitalization. What is the purpose of this statement? The study is for the ICU patients, and I am sure the authors will agree that they don’t have 20-40% admissions without vascular access in their ICU. Please either remove it or provide the appropriate ICU vascular access rate.

Our initial statement was intended to provide broader context about the use of vascular access devices in hospitalized patients. We have revised the sentence. 

“Intravascular devices are placed in the vast majority of critically patients worldwide and up to 80% of critically ill patients have central venous devices [6-8].” Thank you.

[6] Specific to peripheral intravenous lines, removal was also considered a device complication, given the relatively low ‘voluntary removal’ of peripheral venous access in intensive care. What does this statement mean? Should patients stay in ICU when they don’t even need vascular access?

Thank you. ‘Removal’ was used to indicate that the specific peripheral intravenous device being followed had been removed. There are no other intended implications. Accordingly the removal of a PIV does not imply lack of other vascular access nor is it intended to be a statement about a given patients ‘need’ for ICU admission. 

[7] central venous lines(CVL) = let’s call them what they are Central venous catheters or CVCs

This has been revised. Thank you.

[8] The most common complication to occur amongst PIVs was removal. Is removal a complication? Are you thinking about dislodgement?

Thank you. Given our direct observations of the low rate of voluntary removal of (functional) PIV access in our paediatric ICU we have counted all removals as a device complications. In response we have: 

[1] clarified the definition and provided rationale in the methods.

[2] added to the limitations.

We agree that better understanding the various reasons for PIV removal could be important in future work in this and other ICUs. 

7. PLOS authors have the option to publish the peer review history of their article (what does this mean?). If published, this will include your full peer review and any attached files.

Do you want your identity to be public for this peer review? For information about this choice, including consent withdrawal, please see our Privacy Policy.

Reviewer #2: No

Reviewer #3: Yes: Thomas Spentzas

The authors thank the reviewers for their second review and editorial team for their stewardship of this process. 

Thank you for your continued consideration.

C Parshuram and M Gaetani 

For the authors.

---

## [Decision Letter · Decision Letter 2]

19 Jul 2024

PONE-D-23-38349R2Vascular Access Devices and Associated Complications in Paediatric Critical Care: A Prospective Cohort StudyPLOS ONE

Dear Dr. Parshuram,

Thank you for submitting your manuscript to PLOS ONE. After careful consideration, we feel that it has merit but does not fully meet PLOS ONE’s publication criteria as it currently stands. Therefore, we invite you to submit a revised version of the manuscript that addresses the points raised during the review process.

Please revise.

We look forward to receiving your revised manuscript.

Kind regards,

Academic Editor

PLOS ONE

Reviewers' comments:

Reviewer's Responses to Questions

**Comments to the Author**

1. If the authors have adequately addressed your comments raised in a previous round of review and you feel that this manuscript is now acceptable for publication, you may indicate that here to bypass the “Comments to the Author” section, enter your conflict of interest statement in the “Confidential to Editor” section, and submit your "Accept" recommendation.

Reviewer #2: All comments have been addressed

Reviewer #4: (No Response)

2. Is the manuscript technically sound, and do the data support the conclusions?

Reviewer #2: Yes

Reviewer #4: Partly

3. Has the statistical analysis been performed appropriately and rigorously? 

Reviewer #2: N/A

Reviewer #4: Yes

4. Have the authors made all data underlying the findings in their manuscript fully available?

Reviewer #2: Yes

Reviewer #4: Yes

5. Is the manuscript presented in an intelligible fashion and written in standard English?

Reviewer #2: Yes

Reviewer #4: No

6. Review Comments to the Author

Reviewer #2: suitable for publication. All comments are appropriate and all questions addressed. No other concern raised

Reviewer #4: PONE-D-23-38349R2

Vascular Access Devices and Associated Complications in Paediatric Critical Care: A Prospective Cohort Study

Thank you for your submitted manuscript for peer review at PLOS ONE.

This is STROBE-compliant study of pediatric patients who had a vascular access device inserted during a 4.5 yr period in a PICU, with a focus on complications specific to device insertion, lumen occlusion, leaking (which is quite non-specific in its overall definition) and other associated outcomes.

I was not an initial reviewer for the prior revisions until this R2, however I have some concerns regarding the manuscript.

I have highlighted the areas that require further attention before acceptance for publication.

ABSTRACT: overall lacking, however what the conclusion is missing is its summary of the findings - this is usually the first thing I read in an abstract is the conclusion, as this would tell me what the authors outcome recommendations were based upon their study findings - however this does not.

Please avoid bland, generic conclusions like “a wide breadth of device and lumen complications that affect critically ill children. Future work describing strategies that resolve or limit complications are needed to address current knowledge gaps..” - we know that almost all authors make these propositions for future research.. just provide a short but detailed summary of your works findings and save that stuff for the main conclusion.

There is no study timeline mentioned in the abstract methods section, just the facility type. This should be included for clarity. UPDATE: from main manuscript, I found May 2014-Dec 2018 (4.5yrs) - just add 4.5yrs for general study period in the abstract.

KEYWORDS: Please standardize the use of MeSH terms wherever available - details can be found at https://meshb.nlm.nih.gov/

MeSH helps standardize search terms to ensure greater searchability.

Please avoid the use of first-person perspective (we, our, etc.) and use third-person (this study, the authors, this research, these findings, the outcomes, etc.) as this creates a more academically and less personalized styled manuscript.

METHODOLOGY:

Seldinger technique is used for CICC’s - PICCs are inserted using MST (Modified Seldinger Technique) - please correct this oversight so that the correct insertion technique is described for these specific devices.

I think this really should have been an IRB/ethics approved study - regardless of the ‘observational’ aspect of the study, all patients who had a CICC/PICC placed “prospectively” had an invasive procedure performed that would have required written or verbal consent/assent for the procedure - if it was a restrospective study, it would have been different. This is in alignment with any human research as per the Declaration of Helsinki.

RESULTS:

The number of reported devices for a PICU over 4.5 yrs appears to be relatively low in N (220). Is there a reason why? The authors chose to firstly report device days, which can be useful depending on the situation, however it would be worthwhile to report the total number of devices in the study, broken down by each device type.

Reporting generic complication rates is lacklustre. It doesnt truely explain the underlying rationale for these reported complications

Also stating that the most common PIVC complication was removal is useless. Removal is the end-use of the device. It would be more meaningful if the authors stated what was the reason for PIVC removal - accidental/patient initiated, infiltration, extravasation, infection, etc.. these are the underlying reasons for accurate PIVC removal criteria and help define the causal effects.

There is a fair amount of discussion regarding accessory devices (stopcocks, pumps, Y-connectors, etc.) however these are not anlaysed in the tabulated results, which may provide a more impactful understanding of where these complications may be occuring.

4:2:1 rule - what is this? Where are the references that cite this is an empirical protocol that is evidence based? Please provide references to support your statements. This also applies throughout the entire paper as there are other areas that need reference support.

The reference of heparinized saline for PIVCs cites neonatal research, which is a different patient population to pediatrics/children.

I would consider using more age-specific/pediatric research to support your statements.

e.g.

Kleidon TM, Schults J, Rickard CM, Ullman AJ. Techniques and technologies to improve peripheral intravenous catheter outcomes in pediatric patients: systematic review and Meta‐Analysis. Journal of Hospital Medicine. 2021 Dec;16(12):742-50.

Zhang W, Wei B, Chai M, Chen D. Heparin versus normal saline for the care of peripheral intravenous catheters in children: A meta‐analysis. Nursing Open. 2024 Jan;11(1):e2045.

Dobrescu A, Constantin AM, Pinte L, Chapman A, Ratajczak P, Klerings I, Emprechtinger R, Allegranzi B, Zingg W, Grayson ML, Toledo J. Effectiveness and safety of measures to prevent infections and other complications associated with peripheral intravenous catheters: A systematic review and meta-analysis. Clinical Infectious Diseases. 2024 Jun 15;78(6):1640-55.

Indarwati F, Munday J, Keogh S. Peripheral intravenous catheter insertion, maintenance and outcomes in Indonesian paediatric hospital settings: A point prevalence study. Journal of Pediatric Nursing. 2023 Nov 1;73:106-12.

Pacilli M, Bradshaw CJ, Clarke SA. Use of 8-cm 22G-long peripheral cannulas in pediatric patients. The Journal of Vascular Access. 2018 Sep;19(5):496-500.

Please ensure JMIR reporting of P-values in both the abstract and main manuscript, including all tables and figures.

P value is always italicized and capitalized with no spaces between characters.

Do not use 0 before the decimal point for statistical values P, alpha, and beta because they cannot equal 1, in other words, write P<.001 instead of P<0.001

The actual P value* should be expressed (P=.04) rather than expressing a statement of inequality (P<.05), unless P<.001.

P values should not be listed as not significant (NS) since, for meta-analysis, the actual values are important and not providing exact P values is a form of incomplete reporting.

If P>.01 then the P value should always be expressed to 2 digits whether or not it is significant. When rounding, 3 digits is acceptable if rounding would change the significance of a value (eg, you may write P=.049 instead of .05).

If P<.01, it should be expressed to 3 digits.

For P values less than .001, report them as P<.001, instead of the actual exact P value. Expressing P to more than 3 significant digits does not add useful information since precise P values with extreme results are sensitive to biases or departures from the statistical model.

P=.000 (as outputted by some statistical packages) is impossible and should be written as P<.001

This should be in the main body of text as well as all figures/tables where P-values are stated.

Please replace the term “line/lines” throughout the manuscript/tables/figures with the correct terminology/nomenclature ‘catheters’ or ‘devices’ - “line/lines” is an unfortunate frequently used non-descriptive term that is ambiguous in describing venous catheters. Correct nomenclature and terminology are essential in the proper description of medical devices and yet we continue to use/see the term ‘line/lines' to describe various invasive devices; central venous catheters ("central lines”), peripherally inserted central catheters ("PICC lines”), femoral catheters ("femoral lines”), arterial catheters (arterial lines).. a line is nothing but that - a line!

REFERENCES;

20/30 (66%) > 5yrs old with 10/30 (33%) >10yrs old.

I woud caution on the frequent citing/use of older literature, especially from the pre-ultrasound era, as procedural complications have been positively impacted with the upsurge in US use for procedureal safety and procedural efficiencies. I have provided a number of publications that would provide additional support to your submission.

Some further suggested readings;

Nickel B, Gorski L, Kleidon T, Kyes A, DeVries M, Keogh S, Meyer B, Sarver MJ, Crickman R, Ong J, Clare S. Infusion therapy standards of practice. Journal of Infusion Nursing. 2024 Jan 1;47(1S):S1-285. DOI: 10.1097/NAN.0000000000000532

Pittiruti M. Vascular access in pediatric patients: classification and indications. InVascular access in neonates and children 2022 Jun 4 (pp. 3-24). Cham: Springer International Publishing.

Barone G, D’Andrea V, Ancora G, Cresi F, Maggio L, Capasso A, Mastroianni R, Pozzi N, Rodriguez-Perez C, Romitti MG, Tota F. The neonatal DAV-expert algorithm: a GAVeCeLT/GAVePed consensus for the choice of the most appropriate venous access in newborns. European Journal of Pediatrics. 2023 Aug;182(8):3385-95.

Spencer TR, Pittiruti M. Rapid Central Vein Assessment (RaCeVA): a systematic, standardized approach for ultrasound assessment before central venous catheterization. The journal of vascular access. 2019 May;20(3):239-49.

Pittiruti M, Annetta MG, D’andrea V. Point-of-care ultrasound for vascular access in neonates and children. European Journal of Pediatrics. 2024 Mar;183(3):1073-8.

Crocoli A, Martucci C, Persano G, De Pasquale MD, Serra A, Accinni A, Aloi IP, Bertocchini A, Frediani S, Madafferi S, Pardi V. Vascular access in pediatric oncology and hematology: state of the art. Children. 2022 Jan 5;9(1):70.

Georgeades C, Rothstein AE, Plunk MR, Van Arendonk K. Iatrogenic vascular trauma and complications of vascular access in children. InSeminars in Pediatric Surgery 2021 Dec 1 (Vol. 30, No. 6, p. 151122). WB Saunders.

Zito Marinosci G, Biasucci DG, Barone G, D’Andrea V, Elisei D, Iacobone E, La Greca A, Pittiruti M. ECHOTIP-Ped: a structured protocol for ultrasound-based tip navigation and tip location during placement of central venous access devices in pediatric patients. The Journal of Vascular Access. 2023 Jan;24(1):5-13.

Pittiruti M, Celentano D, Barone G, D’Andrea V, Annetta MG, Conti G. A GAVeCeLT bundle for central venous catheterization in neonates and children: a prospective clinical study on 729 cases. The Journal of Vascular Access. 2023 Nov;24(6):1477-88.

Pittiruti M, Crocoli A, Zanaboni C, Annetta MG, Bevilacqua M, Biasucci DG, Celentano D, Cesaro S, Chiaretti A, Disma N, Mancino A. The pediatric DAV-expert algorithm: A GAVeCeLT/GAVePed consensus for the choice of the most appropriate venous access device in children. The Journal of Vascular Access. 2024:11297298241256999.

Annetta MG, Celentano D, Zumstein L, Attinà G, Ruggiero A, Conti G, Pittiruti M. Catheter-related complications in onco-hematologic children: A retrospective clinical study on 227 central venous access devices. The Journal of Vascular Access. 2024 Mar;25(2):512-8.

Paterson RS, Chopra V, Brown E, Kleidon TM, Cooke M, Rickard CM, Bernstein SJ, Ullman AJ. Selection and insertion of vascular access devices in pediatrics: a systematic review. Pediatrics. 2020 Jun 1;145(Supplement_3):S243-68.

Ullman AJ, Gibson V, Takashima MD, Kleidon TM, Schults J, Saiyed M, Cattanach P, Paterson R, Cooke M, Rickard CM, Byrnes J. Pediatric central venous access devices: practice, performance, and costs. Pediatric Research. 2022 Nov;92(5):1381-90.

Kleidon TM, Doellman D, Pitts S, Stranz M. Vascular access by specialists. Pediatrics. 2020 Jun 1;145(Supplement_3):S285-7.

SUPPLEMENTARY DOC:

S1 Table: Outcome definitions - no references to define where these definitions came from. Please address and cite appropriate resources/references.

A number of definitions can be found in the 2024 INS Standards of Practice - Nickel B, Gorski L, Kleidon T, Kyes A, DeVries M, Keogh S, Meyer B, Sarver MJ, Crickman R, Ong J, Clare S. Infusion therapy standards of practice. Journal of Infusion Nursing. 2024 Jan 1;47(1S):S1-285. DOI: 10.1097/NAN.0000000000000532

OVERALL IMPRESSION: I think the authors have a significant amount of revision that is still due with this submission to get it to an acceptable level for publication. I have provided a number of comments regarding specific issues that need to be clearly addressed. While there is no doubt of the importance of the topic, especially in children, this research has been written with amore wideangle lens than drilling down into the cuasal effects of specific complications. I think this would be a more useful process by performing a deep-dive evaluation of the different devices included in the study, rather than just skimming the surface.

7. PLOS authors have the option to publish the peer review history of their article (what does this mean?). If published, this will include your full peer review and any attached files.

Reviewer #2: No

Reviewer #4: No

---

## [Author Response · Author response to Decision Letter 2]

13 Aug 2024

Please see the uploaded response to reviewers document.

---

## [Decision Letter · Decision Letter 3]

16 Aug 2024

Vascular Access Devices and Associated Complications in Paediatric Critical Care: A Prospective Cohort Study

PONE-D-23-38349R3

Dear Dr. Parshuram,

We’re pleased to inform you that your manuscript has been judged scientifically suitable for publication and will be formally accepted for publication once it meets all outstanding technical requirements.

Kind regards,

Academic Editor

PLOS ONE

Additional Editor Comments (optional):

Reviewers' comments:

Reviewer's Responses to Questions

**Comments to the Author**

1. If the authors have adequately addressed your comments raised in a previous round of review and you feel that this manuscript is now acceptable for publication, you may indicate that here to bypass the “Comments to the Author” section, enter your conflict of interest statement in the “Confidential to Editor” section, and submit your "Accept" recommendation.

Reviewer #2: All comments have been addressed

Reviewer #4: All comments have been addressed

2. Is the manuscript technically sound, and do the data support the conclusions?

Reviewer #2: Yes

Reviewer #4: Yes

3. Has the statistical analysis been performed appropriately and rigorously? 

Reviewer #2: Yes

Reviewer #4: Yes

4. Have the authors made all data underlying the findings in their manuscript fully available?

Reviewer #2: Yes

Reviewer #4: Yes

5. Is the manuscript presented in an intelligible fashion and written in standard English?

Reviewer #2: Yes

Reviewer #4: Yes

6. Review Comments to the Author

Reviewer #2: I have no further comment. the manuscript is now suitable for publication on PLOS one after the adequate review made by the Authors

Reviewer #4: Thank you for providing detailed revision as asked during the last peer evaluation period.

This has greatly improved the manuscript overall.

7. PLOS authors have the option to publish the peer review history of their article (what does this mean?). If published, this will include your full peer review and any attached files.

Reviewer #2: **Yes: **Alessandro Crocoli

Reviewer #4: No

---

## [Editor Report · Acceptance letter]

20 Aug 2024

PONE-D-23-38349R3 

PLOS ONE

Dear Dr. Parshuram, 

I'm pleased to inform you that your manuscript has been deemed suitable for publication in PLOS ONE. Congratulations! Your manuscript is now being handed over to our production team.

Kind regards, 

on behalf of

Dr. Robert Jeenchen Chen 

Academic Editor

PLOS ONE